# Homogeneity of agriculture landscape promotes insecticide resistance in the ground beetle *Poecilus cupreus*

**Grzegorz Sowa** [1]*, **Agnieszka J. Bednarska** [2], **Elżbieta Ziółkowska** [1], **Ryszard Laskowski** [1]

**1** Institute of Environmental Sciences, Jagiellonian University, Kraków, Poland, **2** Institute of Nature Conservation, Polish Academy of Sciences, Kraków, Poland

* grzegorz.sowa@uj.edu.pl

**Data Availability Statement:** The raw experiment data set has been deposited in the Zenodo repository under DOI number: https://doi.org/10.5281/zenodo.6384293.

## Abstract

The intensification of agriculture leads to increased pesticide use and significant transformation from small fields towards large-scale monocultures. This may significantly affect populations of non-target arthropods (NTA). We aimed to assess whether the multigenerational exposure to plant protection products has resulted in the evolution of resistance to insecticides in the ground beetle *Poecilus cupreus* originating from different agricultural landscapes. Two contrasting landscapes were selected for the study, one dominated by small and another by large fields. Within each landscape the beetles were collected at nine sites representing range of canola coverage and a variety of habitat types. Part of the collected beetles, after acclimation to laboratory conditions, were tested for sensitivity to Proteus 110 OD–the most commonly used insecticide in the studied landscapes. The rest were bred in the laboratory for two consecutive generations, and part of the beetles from each generation were also tested for sensitivity to selected insecticide. We showed that the beetles inhabiting areas with medium and large share of canola located in the landscape dominated by large fields were less sensitive to the studied insecticide. The persistence of reduced sensitivity to Proteus 110 OD for two consecutive generations indicates that either the beetles have developed resistance to the insecticide or the chronic exposure to pesticides has led to the selection of more resistant individuals naturally present in the studied populations. No increased resistance was found in the beetles from more heterogeneous landscape dominated by small fields, in which spatio-temporal diversity of crops and abundance of small, linear off-crop landscape elements may provide shelter that allows NTAs to survive without developing any, presumably costly, resistance mechanisms.

## Introduction

Agriculture is the most important type of land-use in Europe, covering almost half of the terrestrial area of European Union (EU), and the major pressure on biodiversity and provision of ecosystem services [1]. In agricultural areas important roles are played by all kinds of non-

**Funding:** GS is the author who received each grant. The study was supported by the National Science Centre (https://www.ncn.gov.pl/?language=en), Poland, project Preludium 2016/23/N/NZ8/01679 and Jagiellonian University (https://en.uj.edu.pl/en) in Kraków (DS/WBiNoZ/INoŚ/759/2018). The funders had no role in study design, data collection and analysis, decision to publish, or preparation of the manuscript.

**Competing interests:** The authors have declared that no competing interests exist.

cropped habitats, serving as a reservoir for many plant and animal species [2]. Natural or semi-natural habitats provide not only shelter, reproduction and hibernation sites but also serve as starting points for cyclic recolonization of fields after mowing, fertilization, tillage or pesticide applications and harvest [3,4]. Unfortunately, the intensive development of agriculture promotes large-scale monoculture farming leading to a decrease in spatio-temporal heterogeneity of agricultural landscapes. Additionally, the ever-increasing demand for agricultural products makes it impossible to stop using plant protection products. Although it is difficult to disentangle the impacts of intensified management of local fields from changes in land use at the landscape level–as both occur simultaneously in most agricultural landscapes [5]–one of the most important components of agricultural intensification behind the undesired loses of non-target arthropods (NTA) in agricultural landscapes [6] is the widespread use of pesticides [7]. Pesticides are a quick, highly effective, easy to access, consistent and easy to use tool of choice for farmers for controlling weeds and pests in agriculture landscapes. The worldwide use of agrochemicals has led, however, to contamination of almost every part of our environment in even the most remote places [8]. Because of the similarity of basic biochemical processes among insects and many other invertebrates, insecticides are not specific enough to affect only pest species, and have harmful effects also on beneficial organisms inhabiting agricultural landscapes [7]. The eventual effect of insecticides on NTA depends on which specific products are used (some are more toxic and/or persistent than others), how they are used (e.g., repetitive use may lead to the accumulation of a chemical and/or its effects but also to the development of resistance in chronically exposed populations), and what is the environmental context (e.g., landscape structure).

Carabids, as a taxonomically and ecologically diverse group, have different habitat requirements and may respond differently to landscape structure and its management [9–11]. Both larvae and adults of most carabid species are carnivorous and are known to be predators of many invertebrate pests, but can also consume seeds of different weeds [12,13]. Virtually all agriculture practices result in direct or indirect effects on carabid communities, either through mortality and emigration or changes in conditions in occupied microhabitats [14,15]. Carabids are sensitive to many insecticides, including all the most commonly used groups: organophosphates, organochlorines, carbamates, pyrethroids, and neonicotinoids [16]. During and/or soon after a spraying, adult beetles, which spend most of the time at the soil surface when not hibernating, may easily be exposed topically to pesticide droplets from a sprayer and/or falling from sprayed plants. Other exposure routes include consumption of contaminated prey and seeds, contact with contaminated soil, water, and plant surfaces.

In the field, apart from direct mortality caused by insecticides, NTAs, including carabids, are frequently exposed to sub-lethal doses that can lead to more subtle effects, for example on learning performance, behavior and neurophysiology [17]. Organisms chronically exposed to low doses of pesticides may not show signs of acute toxicity (e.g., in the form of increased mortality), but may have reduced tolerance to other stress factors such as other toxic chemicals and/or various types of natural environmental factors, e.g., low or high temperatures, shortage of food, etc. [18]. To survive in environments with a long history of intensive agriculture, where insecticides have been used regularly for many years, populations of any species may be pushed towards evolution of resistance to the most commonly used plant protection products. Although this phenomenon has been proved for many pest populations [19–21], it is much less recognized in NTAs [22–25]. The evolution of resistance, although temporarily beneficial to organisms living in an insecticide -treated area, may be metabolically costly [26], making insecticide-resistant individuals less fit in uncontaminated environments and/or more sensitive to other stressors [18]. Hence, even if increased resistance among NTAs may seem beneficial, it may bring undesired long-term effects. On the other hand, the presence of resistance in

pest insects may lead to increased use of insecticides [27]. This potentially may start a cascade of unwanted effects, jeopardizing numerous NTA populations and even bringing human health hazard. Maintaining NTAs biodiversity, abundance and stable populations is, thus, important also from an economic and human health perspective.

It may be expected that changes caused by chronic exposure to pesticides in the field are expressed to a different degree in populations inhabiting differently transformed agricultural areas. Therefore, understanding changes in NTA communities along gradients of agricultural landscape complexity will help us to not only sustain vital habitat conditions to prevent beneficial species from extinction, but also to save billions of dollars which are spent on plant protection products and practices [28]. In the end, understanding insect responses to pesticide pressures in their local ecological context poses a key challenge in developing balanced pest control strategies.

The objective of this study was to investigate the importance of agricultural landscape structure to the sensitivity of the carabid beetle *Poecilus cupreus*, representing an important group of NTAs–the pest control agents, to insecticides and the evolution of resistance in its populations. We located our study area in the agricultural landscapes of the Wielkopolska [Greater Poland] province in Poland where, within a relatively small area, hugely different landscapes can be found, ranging from large-scale agriculture to more traditional small-scale farming. We hypothesized that populations inhabiting agricultural landscape dominated by large fields with high percentage of canola coverage exhibit increased resistance to the insecticide most commonly used in the area (Proteus 110 OD), while small-fields landscape, with its high diversity of crops and, especially, off-crop habitats serving as refuge areas (e.g., field margins), prevents development of resistance in beetles. We focused in our study on the canola (winter variety) because it is an important crop in the European Union (10% of the EU's arable land). Although the production of canola in EU is slowly decreasing, in Poland we can still see an increase in the production [29]. Canola plantations usually receive one to four insecticide applications in a season, but in some cases there can be even five or more treatments [30]. We assumed, thus, that the gradient of canola coverage represents a gradient of pesticide pressure. Most treatments in canola plantations are carried out in spring and early summer because this is when the risk of pest outbreaks is the highest. Adult *P. cupreus*, as spring breeders, are present in the fields during spraying, and therefore can be particularly exposed to plant protection products via direct spray or residues on plants, food, water and soil, possibly favoring the evolution of resistance towards insecticides. To sort out possible temporary effects through direct selection of the most resistant individuals collected from the field right after the spraying from possible genetically fixed adaptation to the used pesticides, the sensitivity of beetles towards Proteus 110 OD was tested on field collected beetles (P) and two consecutive laboratory cultured generations (F1 and F2).

## Methods

### Study area and site selection

The study area lies in the southwest part of the Wielkopolska province (western Poland, Fig 1) and represents a typical farmland, where arable fields occupy nearly 65–70% of the area. The growing season is one of the longest in Poland, beginning around the end of March and lasting for approximately 220 days. Mean annual temperature is ca. 8˚C and mean annual precipitation over 550 mm [31,32]. Within the study area, two distinct landscapes, 12 x 16 km each, were selected: one dominated by large fields (hereafter 'large-fields landscape', L) and the second one with prevailing small-fields family farming (hereafter 'small-fields landscape', S) (Fig 1). Both landscapes are similar in terms of land cover (i.e., percentage of arable areas,

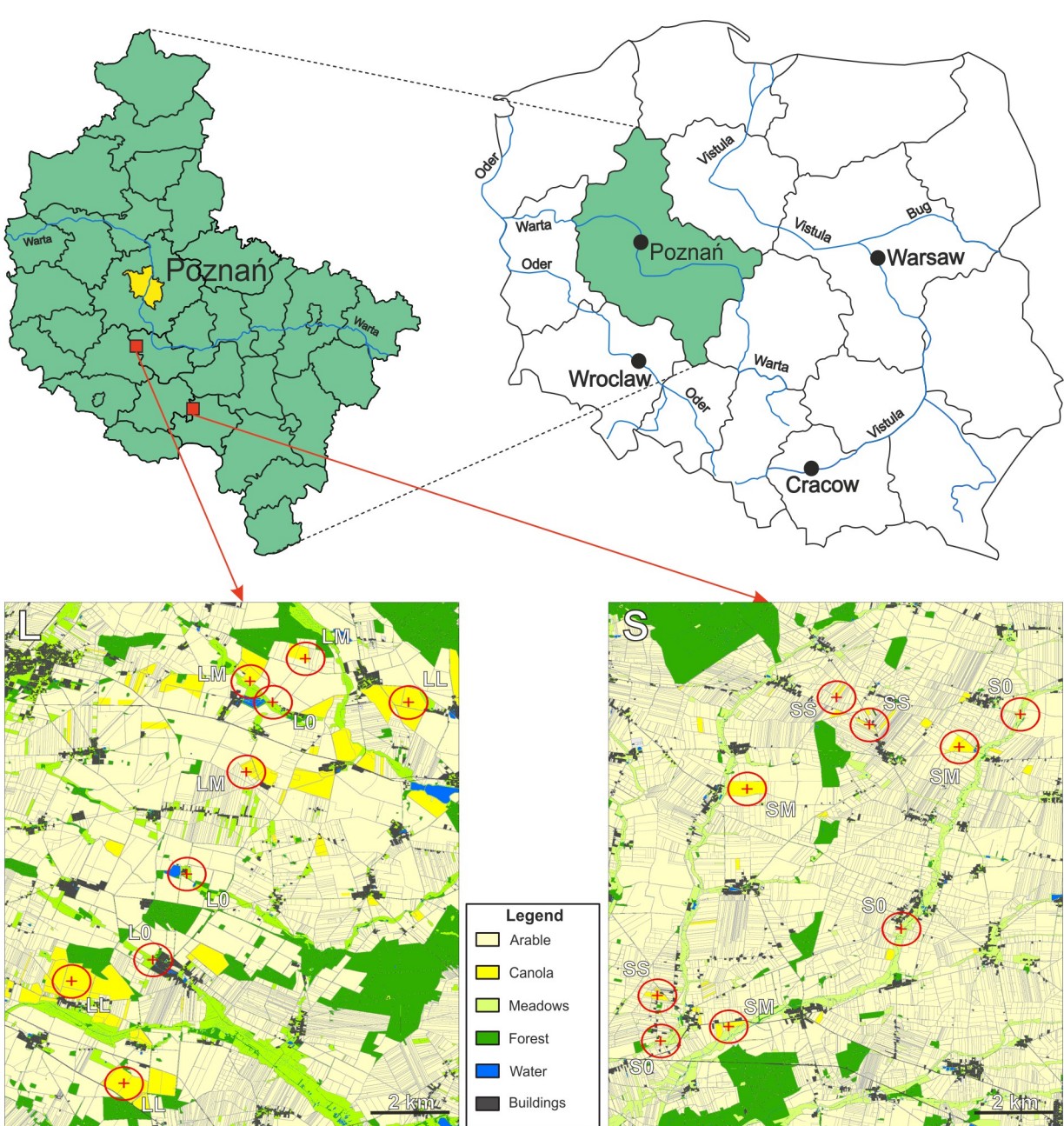

**Fig 1. Location of the study landscapes and sites.** Upper right, administrative division of Poland into provinces. In green–Wielkopolska province. Upper left–administrative division of Wielkopolska with the capital of the region [Poznań] marked in yellow and the location of the two study landscapes (red squares; enlarged bellow) representing two contrasting landscape types: S-small-fields landscape, L-large-fields landscape. Within each landscape the locations of the study sites are marked (red circles– 500 m radius). The first character stands for landscape type (S vs. L); the second character stands for canola coverage: 0 –none, S–small (10–14%), M–medium (20–52%), L–large (80–98%). Land-cover maps were generated according to methodology described by Ziółkowska et al. [33], see Methods section for more details.

woodlands, water bodies) and pedo-climatic conditions, but differ in the farmland structure (Table 1).

Raster land-cover maps (resolution of 1 m$^2$) for the landscapes were generated in a step-by-step process according to the methodology described in [33] by combining data from: (1) the National Database of Topographic Objects (BDOT) providing the land cover and land use

**Table 1. Two study landscapes, each 12 x 16 km, with share of land of the main land cover types [%], share of arable land in given field size classes [%], field border length [km] and total number of arable fields.**

|  |  | Large-fields landscape (L) | Small-fields landscape (S) |
|---|---|---|---|
| Share of land cover units [%] in a landscape | Arable | 73.2 | 76.9 |
|  | Herbaceous | 8.8 | 8.2 |
|  | Woodland | 11.2 | 8.7 |
|  | Build up | 5.3 | 4.7 |
|  | Water bodies | 0.8 | 0.6 |
|  | Other | 0.7 | 0.9 |
| Share of arable land [%] in given field size classes | < 3 ha | 23.5 | 41.7 |
|  | 3–10 ha | 25.7 | 40.1 |
|  | 10–30 ha | 21.7 | 11.5 |
|  | 30–50 ha | 16.8 | 4.8 |
|  | ≥ 50 ha | 12.3 | 1.9 |
| field borders [km] |  | 4443 | 6131 |
| total number of arable fields |  | 4072 | 6494 |

information at the scale of 1:10.000, and (2) the Land Parcel Identification System (LPIS) providing information on the type of cultivated crops. In Poland, LPIS, managed by the Polish Agency of Restructuring and Modernization of Agriculture, is based on the national land and building cadaster and therefore information on type of cultivated crop is provided at the level of cadastral parcels. From LPIS we used information for 2018 to allocate major crop types to individual cadastral parcels. The information for canola was further verified in the field, together with delineation of individual fields if the cadastral parcel was shared by more crops than canola only. Spatial data were handled and analyzed using ArcGIS 10.4 (ESRI, Inc., USA).

Based on the analysis of land-cover maps, three habitat types per landscape–each consisting of three study sites (therefore in total nine study sites per landscape)–were selected based on the canola coverage (CC) expressed as percent of total area within a 500 m radius around the midpoint where the beetle traps were located (Fig 1 and Table 2). The midpoints of the study sites with canola were located in canola fields and were separated from each other by at least 800 m to avoid situations were beetles from one population could be caught in different study sites. At the same time, the most distant study sites were located not more than 60 km apart to ascertain similar climatic and edaphic conditions. In the large-fields landscape (L) sites representing the following beetle habitat types were distinguished: with medium CC (28–33%, LM), with large CC (80–98%, LL), and without CC (L0). In the small-fields landscape (S), it was impossible to establish study sites with CC larger than 60% due to the lack of large canola fields and therefore the beetle habitat types were distinguished as follows: with small CC (10–14%, SS), with medium CC (20–52%, SM), and without CC (S0). Note that in both landscapes, sites with no CC and with medium CC were established and could serve for direct comparison between the landscapes. The midpoints (beetle traps) of study sites L0 and S0, serving as a control, were located on meadows subjected to agricultural practices with the exclusion of pesticide applications. The whole area, in both landscapes, is strongly dominated by conventional agriculture, and no fields managed as organic were present in our study areas.

## Study species

The ground beetle *Poecilus cupreus* was chosen for investigation because it was numerous enough at all the study sites, easy to identify in the field, and with a well-tested culturing procedure. It is one of the most common and dominant carabids found on arable land across

**Table 2. Geographic coordinates (decimal degrees) of the study sites.**

| Habitat type | Coordinates | |
|---|---|---|
| | Longitude | Latitude |
| L0 | 52.1341 | 16.8568 |
| L0 | 52.0627 | 16.8140 |
| L0 | 52.0864 | 16.8559 |
| LM | 52.1462 | 16.8688 |
| LM | 52.1396 | 16.8478 |
| LM | 52.1148 | 16.8475 |
| LL | 52.1349 | 16.9091 |
| LL | 52.0563 | 16.7829 |
| LL | 52.0287 | 16.8044 |
| S0 | 51.8162 | 17.3959 |
| S0 | 51.7568 | 17.3524 |
| S0 | 51.7249 | 17.2615 |
| SS | 51.8198 | 17.3252 |
| SS | 51.8125 | 17.3381 |
| SS | 51.7373 | 17.2597 |
| SM | 51.8069 | 17.3728 |
| SM | 51.7943 | 17.2919 |
| SM | 51.7292 | 17.2873 |

Europe [34] and an example of a typical beneficial predator [35]. Adult beetles are strictly diurnal and disperse mainly by walking but can occasionally fly [36]. Usually individuals do not show a high level of movement throughout the active period that falls in spring–summer. It was demonstrated that beetles disperse over hundreds of meters depending on landscape composition and availability of resources [37]. *Poecilus cupreus* is associated with many different crops, with canola being the most favorable spring habitat [38]. It also inhabits different types of meadows with relatively high soil humidity [39]. *Poecilus cupreus* is a typical representative of so-called spring breeders, with the period from April to the end of July being the main reproductive time. New generation adults emerge in August, and in the late September the beetles start overwintering hibernation [40]. As *P. cupreus* can live up to 2–3 years, it is possible for adults to have two activity periods: the first, shortly after hatching just before winter diapause and the second, longer one, in the following spring and summer.

## Beetle collection and laboratory culture

In the midpoint of each study site, 64 Barber traps without any preservative were set up in a grid of 8 x 8 m (64 m$^2$). The beetles were collected in 2018 during their peak activity in April–May. The traps were emptied every 2–3 days, the beetles sorted in the field and individuals of *P. cupreus* were placed in plastic containers (23 x 17 x 11 cm) with moist peat, transported to the laboratory and kept in a climatic chamber (relativity humidity 70% ± 5%, temperature 20°C ± 2°C; day:night 16 h:8 h). The authorisation for capture, transport and keeping the beetles was granted by the Regional Directorate for Environmental Protection in Kraków, Poland, document no. OP-I.6401.128.2017.MMr, and by the Regional Directorate for Environmental Protection in Poznań, Poland, document no. WPN-II.6401.83.2017.AG.2.

To obtain sufficient numbers of beetles for each habitat type, the beetles from three sites representing particular CC within particular landscape type were pooled together. Part of the field collected beetles (P generation) were utilized for insecticide sensitivity test on parental

generation and the remaining beetles were used to establish laboratory cultures to obtain the next two generations (F1 and F2). The beetles were cultured according to Bednarska and Laskowski [41] procedure with only one modification: the animals were fed *ad libitum* with artificial food made of frozen ground mealworms mixed with ground apple but without any preservative to eliminate contact with any potentially harmful chemicals. After overwintering, adult beetles from the laboratory cultures (F1 and F2 generations) underwent the same experimental procedure as the parental generation.

## Experimental design

In total, 250–280 individuals were tested for sensitivity to the insecticide in each generation, with 15–40 beetles per habitat type per landscape, depending on availability. Proteus 110 OD (Bayer, Germany) was selected for the experiment as the most frequently used insecticides in the studied landscapes (according to the survey on pesticide usage conducted among local farmers and farmer advisors from regional agricultural advisory center), applied only in winter rape in the period of sampling the beetles (April-May). It has two active ingredients: thiaclo-prid (100 g $L^{-1}$) and deltamethrin (10 g $L^{-1}$) (see S1 Table in Supplementary materials for full product specification). Nevertheless for testing the resistance of beetles to insecticides other formulations might as well have been used, as long as their active ingredients belong to the main classes of insecticides (here, pyrethroids and neonicotinoids) commonly used in the studied landscapes. If some resistance mechanisms become established in a population, they are universal for all insecticides of a given class, not only for a specific chemical compound representing that class.

Twenty-four hours before insecticide application, the beetles were placed individually in plastic Petri dishes with a diameter of 35 mm (FL Medical, Italy) to acclimatize. The commercial formulation of Proteus 110 OD was dissolved in acetone to obtain the concentration equivalent to 0.2 recommended concentration for field use (for canola pests 0.6 L of the product in 300 L of water per hectare is recommended). The beetles from each habitat type were exposed individually to a single topical application of 1 μl droplet of the insecticide solution applied on the scutellum with a Hamilton syringe with a repeater (Hamilton Company, USA). For control habitats (S0 and L0), additional 30–40 individuals per habitat were treated with 1 μl of pure acetone only (these groups are denoted S0-A and L0-A, respectively) to confirm the effectiveness of the insecticide dose used (positive control). Because S0 and L0 populations did not differ in survival within treatment groups (i.e., either acetone- or Proteus-treated), the S0 and L0 beetles from a given treatment were pooled together and tested for the insecticide effect by comparing the survival curves.

After the treatment, the beetles were placed back in the climatic chamber. The mortality and immobility were recorded after 0.5, 1, 2, 3, 6, 8, 10, 12, 24 h and daily afterwards. In the analysis of mortality, we assumed "dead" to be all individuals that either indeed died or were paralyzed (unable to walk/relocate). The beetles were not fed during the experiment. Each experiment (i.e., on generations P, F1 and F2) was terminated when all the individuals had died. The dead beetles were placed into 2 ml Eppendorf tubes (Sarstedt AG and Co. KG, Germany) and dried at 105˚C for 24 h to obtain dry mass. We weighed all the beetles in order to exclude body mass as a factor for possible differences in susceptibility to the insecticide. Dry body mass was recorded to the nearest 0.00001 g using XA 110/2X scale (Radwag, Poland).

## Statistical analysis

Survival curves of beetles originating from different habitat types were compared using survival analysis (Wilcoxon test), assuming $p \leq 0.05$ as statistically significant. We choose

Wilcoxon test because it is more suitable than log-rank test when the hazard is higher at early survival times than later [42] as is the case for insecticide-treated organisms [43]. The survival analyses were performed separately for each generation. First, we compared survival curves of beetles from habitat types with no CC, but separately for insecticide-treated (S0 and L0) and acetone-treated (S0-A and L0-A) ones. Subsequently, survival curves for the insecticide-treated beetles within a particular generation were analyzed for the effect of habitat type. As statistically significant differences were found, this was followed by pairwise comparisons of each habitat type against one another. Dry body mass of beetles was compared among all habitat types and generations using two-way analysis of variance with habitat type and generation as factors. Because both factors and the interaction were highly significant (p ≤ 0.0002), in the next step we performed one-way ANOVA with Tukey post hoc test separately for each generation. All statistical analyses were performed using Statgraphics Centurion 18 (Statgraphics Technologies, Inc., USA).

## Results

### Parental generation (P–Field collected beetles)

The survival curves for beetles from control habitats from the two analyzed landscapes did not differ, regardless if beetles were treated with insecticide (S0 vs. L0, p = 0.95) or acetone only (S0-A vs. L0-A, p = 0.52, Fig 2). The insecticide significantly increased the mortality rate of the meadow beetles (p < 0.0001 for all three generations), confirming that the insecticide dose used was effective. Thus, meadow populations of beetles from S and L landscapes were similar in terms of their background mortality (with no insecticide treatment) and in terms of their sensitivity to the insecticide. Wilcoxon test showed, however, a significant (p ≤ 0.0001) difference in survival rates of insecticide-treated beetles originating from different habitat types (Fig 3). Pairwise comparisons of the habitats with one another revealed that only insecticide-treated beetles from the LM and LL habitats survived significantly better than those from other habitat types (Fig 3, Table 3), and even from acetone-treated beetles from the control habitats (S0-A and L0-A). The LT50 values for beetles from the LM habitat type were almost three times higher than in beetles from the S0 and SM habitat and ca., 1.5 times higher than those found for beetles from the control habitats treated with acetone only (S0-A and L0-A) (Fig 4; Table 3). Although SM beetles originated from habitats similar to LM in terms of CC, they had the lowest LT50 recorded for P generation.

The one-way ANOVA revealed a statistically significant difference in beetle dry body mass among habitat types (p ≤ 0.0001). The dry mass of both insecticide-treated and acetone-treated beetles from the control habitats in large-fields landscape (L0 and L0-A) was the lowest, and significantly lower than in all other habitat types with the only exception that L0 did not differ from S0 (Fig 5A).

### Laboratory bred generations (F1 and F2)

As in the case of P generation, the survival curves of F1 and F2 generations of beetles originating from control habitats showed no statistically significant difference either for acetone-treated beetles (S0-A vs. L0-A, p = 0.35 for F1, p = 0.85 for F2, Fig 2) or insecticide-treated beetles (S0 vs. L0, p = 0.12 for F1, p = 0.17 for F2, Fig 2). Also as in the P generation, the comparison among insecticide-treated beetles from different habitat types revealed statistically significant difference in survival rates of beetles in both the F1 and F2 generations (p < 0.001 for F1, p < 0.03 for F2, Fig 3). The increased tolerance to the insecticide found in beetles from the P generation from the LM and LL habitats was still present in both next generations F1 and F2, as insecticide-treated beetles from the LM and LL habitats survived significantly better

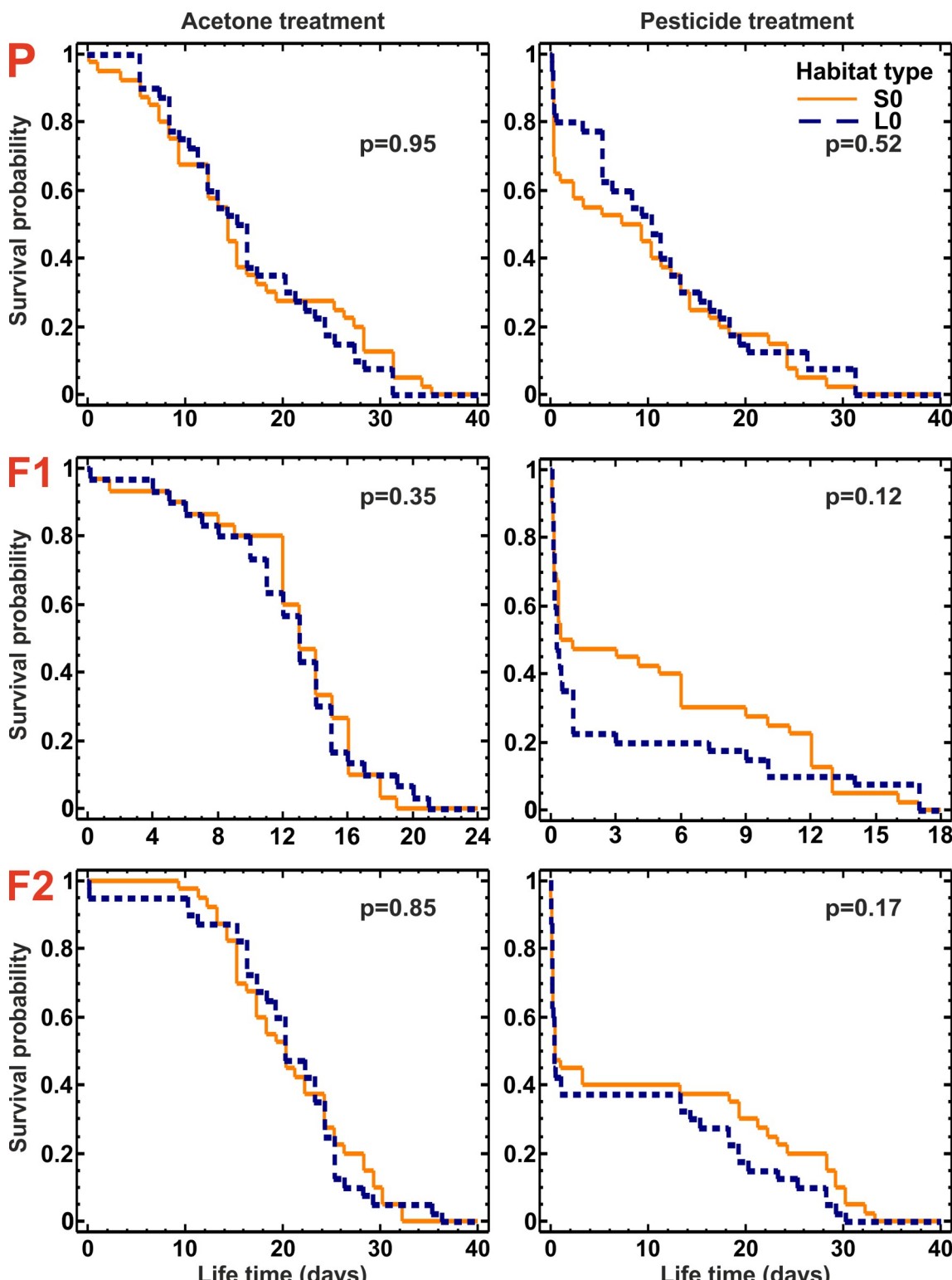

**Fig 2. Survival curves of all three generations of *Poecilus cupreus* from control habitats after exposure to acetone and Proteus 110 OD pesticide treatment.** Solid lines stand for small-fields landscape; dashed lines stand for large-fields landscape. The first character stands for landscape type: S–small fields landscape, L–large fields landscape; the second character stands for canola coverage: 0 –none. Red uppercase letters indicate generations: P–parental, F1 & F2 –laboratory breed. In all three generations the insecticide treatment significantly increased the mortality rate (p < 0.0001); p values on the graphs indicate the significance level for comparisons between S0 and L0 beetles.

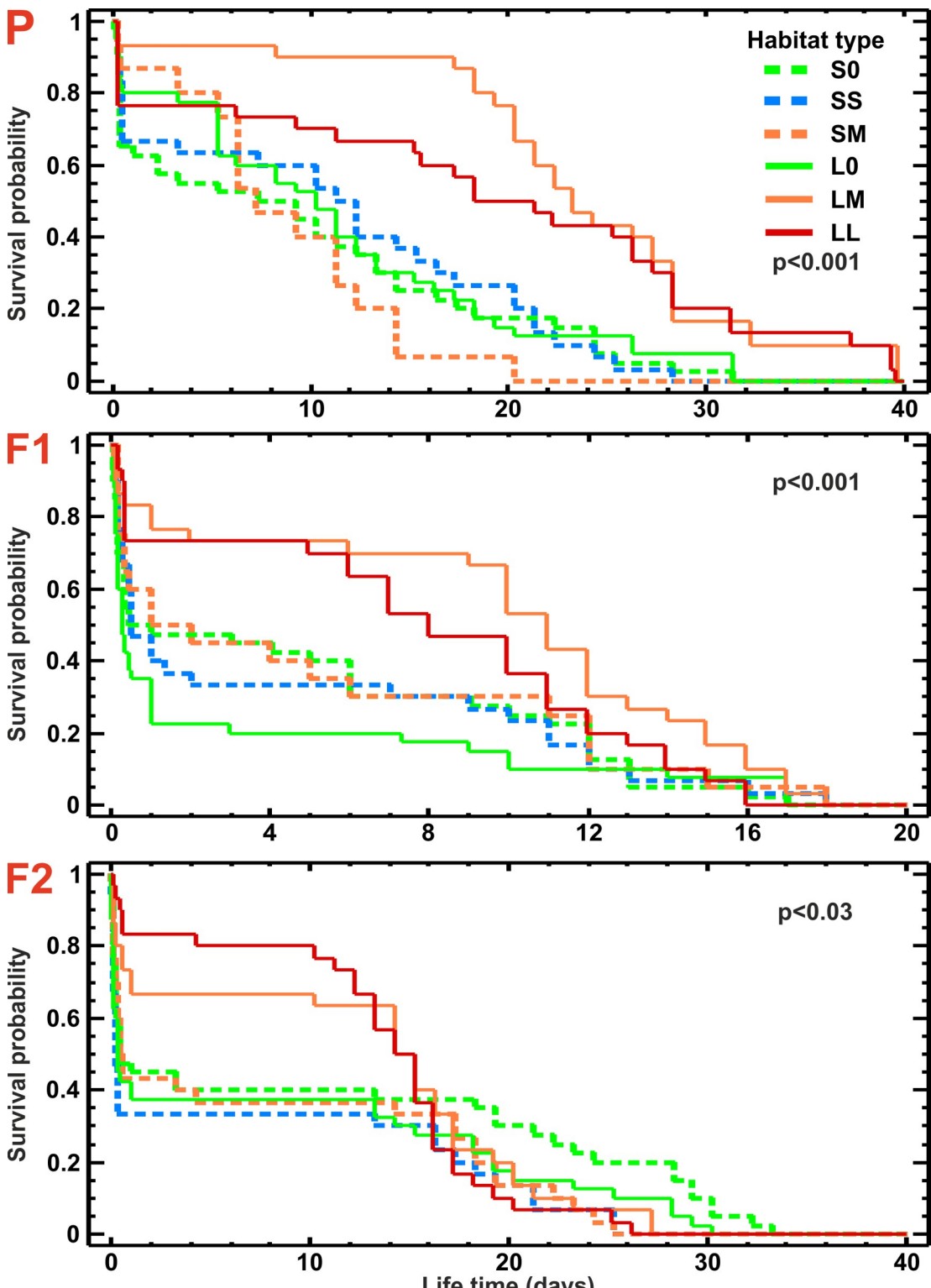

**Fig 3. Survival curves of all three generations of *Poecilus cupreus* from different habitat types and landscapes after exposure to the insecticide Proteus 110 OD.** Dashed lines stand for small-scale landscape; solid lines stands for large-scale landscape. The first character stands for landscape type: S–small-fields landscape, L–large-fields landscape; the second character stands for canola coverage: 0 –none, S–small (10–14%), M–medium (20–52%), L–large (80–98%). Red uppercase letters indicate generations: P–parental, F1 & F2 –laboratory breed. Note the generally shorter survival times in the first laboratory generation (F1).

**Table 3. Median lethal times (LT50, days) for all three generations (P–Parental, F1, F2 –Laboratory bred) from all habitat types.**

| Site | LT50 (days) | | |
|------|-------------|---|---|
| | **P** | **F1** | **F2** |
| *S0-A* | *14.32[b] (40)* | *13.03[d] (30)* | *20.29[d] (40)* |
| S0 | 7.31[a] (40) | 0.42[ab] (40) | 0.42[abc] (40) |
| SS | 12.3[a] (30) | 0.5[ab] (30) | 0.25[a] (30) |
| SM | 7.29[a] (15) | 1[bc] (20) | 0.5[ab] (30) |
| *L0-A* | *15.28[b] (40)* | *13[d] (30)* | *20.26[d] (40)* |
| L0 | 10.27[a] (40) | 0.25[a] (40) | 0.33[a] (40) |
| LM | **23.26[c] (30)** | **10.97[c] (30)** | **15.24[bc] (30)** |
| LL | **21.25[bc] (30)** | **7.96[c] (30)** | **15.23[c] (30)** |

The sites are named as follows: First character stands for landscape type (S–small-fields agriculture, L–large-fields agriculture); second character stands for canola coverage (0 –none, S–small, 10–14%, M–medium, 20–52%, L–large, 80–98%). All groups except those marked with letter "A" (for acetone only) were exposed to the insecticide. Groups with the same lowercase letter do not differ significantly at p ≤ 0.05 in terms of survival (pairwise comparison of survival curves, Wilcoxon test); two groups showing resistance are typed in boldface; beetles originating from meadows and not treated with the insecticide are in italics. The numbers of individuals used in particular treatments are reported in brackets.

than insecticide-treated beetles from all other habitat types (Fig 3, Table 3), but significantly worse than acetone-treated beetles from the control habitats (S0-A and L0-A; Table 3). The LT50s for beetles from the LM and LL habitats, although lower than in the P generation, also in the F1 and F2 were higher than for insecticide-treated beetles from other habitat types (Fig 4).

Although one-way ANOVA on beetle dry body mass revealed statistically significant differences among beetles from different habitat types in both the F1 and F2 generations (p ≤ 0.0001), the beetles from the LM and LL habitats–with confirmed increased tolerance to the insecticide–did not differ significantly in weight from the beetles from most other habitats. The only exception were the F2 beetles from the LM and LL habitats, which were significantly heavier than the F2 beetles from the control habitats, either insecticide- or acetone-treated (L0 and L0-A, Fig 5A). In general, the mean dry body mass of the insecticide-treated beetles from the control habitat in the large-fields landscape (L0) was the lowest in all generations (Fig 5A). Additionally, the dry mass of beetles decreased over the generations (p ≤ 0.0001, Fig 5B), showing that laboratory conditions could have an unforeseen impact on long-term experiments.

## Discussion

The goal of this study was to contribute to the knowledge on the evolution/selection towards elevated insecticide resistance in carabids by investigating the sensitivity of the predatory *P. cupreus* to Proteus 110 OD over three generations. Because we assumed that the development of resistance may depend on the environmental context, we used beetles originating from different habitat types, defined based on canola coverage within the presumed beetle home range, located within two landscapes with distinct farmland heterogeneity. Our results showed that beetles from the landscape dominated by large fields and living in habitats with medium and large coverage of canola (LM and LL habitats) exhibited significantly lower mortality after exposure to Proteus 110 OD than all other populations, and that the difference was maintained in two consecutive laboratory-bred generations. Proteus 110 OD is the plant protection

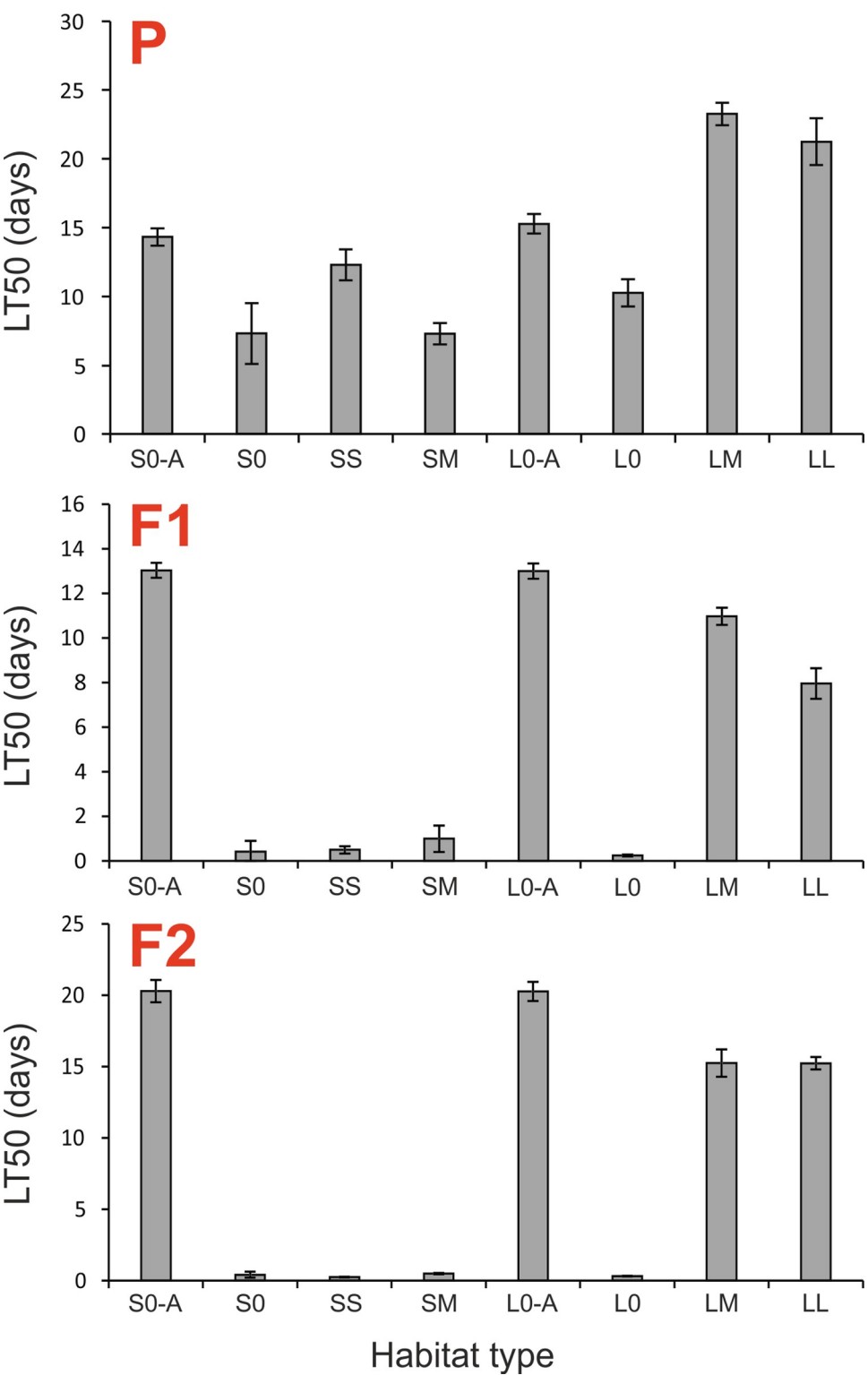

**Fig 4. Median lethal times (LT50s) with standard errors for the three generations of *Poecilus cupreus* from all experimental groups, representing different habitat types, landscapes and treatments.** The first character stands for landscape type (S–small fields landscape, L–large fields landscape); the second character for canola coverage (0 –none, S–small, 10–14%, M–medium, 20–52%, L–large, 80–98%). All groups except those marked with letter "A" were exposed to Proteus 110 OD insecticide. Red uppercase letters indicate generations: P–parental, F1 & F2 –laboratory breed.

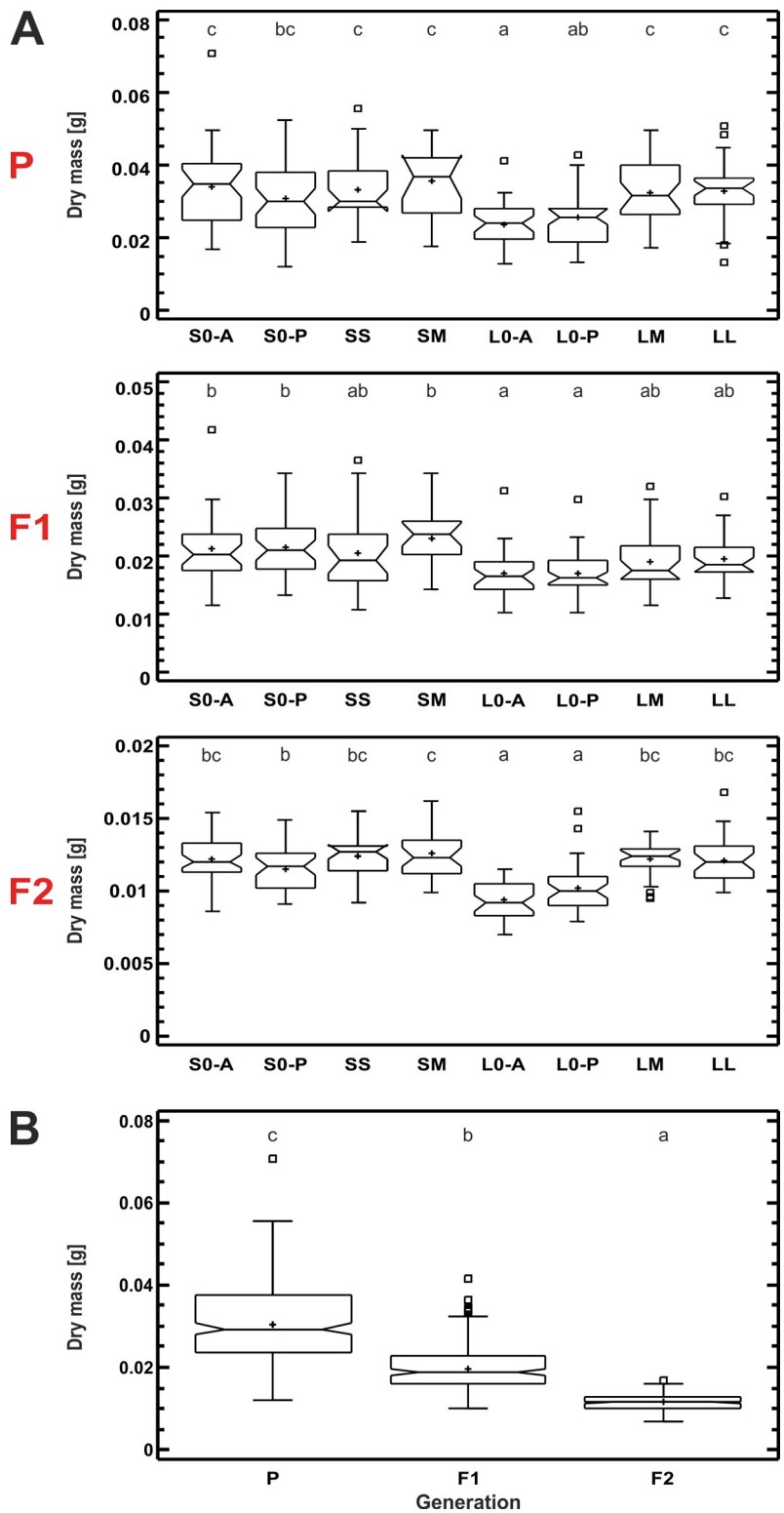

**Fig 5.** Dry body mass in the three generations of *Poecilus cupreus* from all experimental groups, representing different habitat types and treatments (A) and across all generations (B). The first character stands for landscape type: S–small fields landscape, L–large fields landscape; the second character stands for canola coverage: 0 –none, S–small (10–14%), M–medium (20–52%), L–large (80–98%). Within each generation all groups except those marked with letter "A" were exposed to Proteus 110 OD insecticide. Red uppercase letters indicate generations: P–parental, F1 & F2 –laboratory

breed. Means of groups marked with the same letter above box-and-whisker plots do not differ significantly at
$p \leq 0.05$. The graphs indicate median (shorter horizontal line), average (plus sign), second and third quartile (wider
horizontal lines), minimum and maximum (whiskers) except for outliers > 1.5 interquartile rage (asterisks); the notch
indicates approximate 95% confidence interval for the median.

product containing two active ingredients, thiacloprid which belongs to neonicotinoids and
affects insect nervous system by stimulating nicotinic acetylcholine receptors, and deltame-
thrin which is a pyrethroid preventing the closure of the voltage-gated sodium channels in the
axonal membranes resulting in dysfunction of spiracles–insect death is caused by overdrying.
However, the ultimate effect of commercial formulations depends not only on the active ingre-
dients but also adjuvants/co-formulants, which may affect the efficacy of penetration of the
active ingredients into insect bodies, the persistence of an insecticide in field conditions [44].
Hence, other formulations with exactly same active ingredients may have somewhat different
effects than those reported herein.

As all beetles in the experiment received an identical dose of the insecticide (except those
treated with acetone only), theoretically the observed higher survival of the beetles from the
LM and LL habitats could result from higher body mass of the beetles from these two popula-
tions rather than from increased resistance. This hypothesis can be, however, rejected as the
dry mass of the beetles from the LM and LL habitats did not stand out in any way. We can thus
conclude that body mass was not the factor differentiating the beetles in terms of survival after
the insecticide treatment, and that the increased resistance found in populations from the LM
and LL habitats resulted from the few co-varying factors occurring at local and landscape
scales: the presence of large monocultures, homogenization of landscape by disappearance of
non-cultivated elements and/or repeated episodes of pesticide sprays; possibly an increased
abundance of pesticide adapted prey can also be a factor [25].

One of the main mechanisms of developing resistance to an insecticide is detoxification
through the overexpression of metabolic genes. Increased detoxification usually results in
energy reallocation at the expense of metabolic and developmental processes. Because resis-
tance to pesticides seems to be costly and in pesticide-free environments the selection acts
against it [45,46], the increased resistance has low chances to be fixed in populations with low
or sporadic pesticide exposure. Hence, a genetically fixed resistance may be expected in popu-
lations that are chronically and repetitively exposed to insecticides, being thus under continu-
ous strong selective pressure. The exposure to insecticides not only depends on management
of fields at a local scale (the type and number of product applications on a given field) but also
on the context of agriculture and land-use at the landscape level [6]. The necessity to analyze
these mechanisms at various spatial scales simultaneously is clearly visible in our results, as the
increased resistance to Proteus 110 OD was found exclusively in populations from habitats
with medium and large coverage of canola (local-scale measure of stressor's intensity) only if
they were located in landscapes dominated by large fields (landscape-scale measure of stress-
or's intensity).

Large coverage of canola within a local habitat increases the probability of beetles being
directly exposed to repeated episodes of insecticide sprays. It has been shown by Schoonees
and Giliomee [47] that insects receiving more sprays per year exhibited higher resistance to
the used pesticides than those which experienced only one application. However, we showed
that the proportion of canola coverage within a local habitat is not the sole deciding factor for
the evolution of resistance. Beetles originating from SM habitat type did not develop resis-
tance, although the canola coverage in the SM habitat type was similar to the LM habitat type,
indicating the importance of factors acting at a landscape-scale. In landscapes dominated by

large fields, carabids have less possibilities to migrate to non-crop habitats or to fields with other crops where other management practices are used. This not only increases the probability of beetles being exposed to direct spraying but also to contaminated prey, which has been indicated as an important factor contributing to the development of resistance in beneficial organisms [25]. In contrast to the beetles inhabiting landscapes dominated by large fields, those living in more diverse landscapes–like the beetles from SS and SM habitats–can more easily emigrate during spraying or right after to adjacent fields or non-crop habitats and part of the population is always present in unsprayed non-crop areas [48]. Even if pesticide application in canola fields in S landscape is similar to that in L landscape and the temporal selection pressure on beetle populations is similar across all canola fields, the presence of other habitats within the beetle home range in S landscape acts against fixing the resistance genes in local populations.

The necessity of taking into account not only local but rather various spatial scales simultaneously when investigating the impacts of stressors on species in agroecosystems is generally well recognized [6]. Riggi et al. [49] investigated the importance of the proportion of canola coverage and crop heterogeneity at two spatial scales (defined either as a 3 km radius around the sampled canola field or Sweden's administrative region) on the occurrence of resistance to lambda-cyhalothrin insecticide in the pollen beetle–one of the major canola pests. They found a positive effect of the canola coverage on insecticide resistance in pollen beetle populations, but only at the regional spatial scale and suggested an effect of regional landscape history on the current pest resistance. To the best of our knowledge, similar studies on carabid beetles, being natural pest enemies, do not exist. Our results, by showing that heterogeneity at the landscape level can play a major role in the development of increased insecticide resistance, confirm the importance of maintaining or increasing landscape (spatio-temporal) heterogeneity as a mechanism of supporting beneficial organisms in agroecosystems. The presence of extensively managed semi-natural habitats–even if only as narrow, linear elements–plays a key role for NTAs in reducing the negative impacts of insecticides by providing shelter and access to uncontaminated food [48], but also as overwintering refugia or source habitats for the recolonization of fields [7,50]. With landscape simplification and land consolidation, the abundance of semi-natural habitats is decreasing together with the multiple ecosystem services they are providing [51].

The results for dry mass of beetles showed that with each generation the beetles became lighter. One potential explanation may be the inbreeding. Limited gene flow and accumulation of specific mutations in a laboratory culture can result in many changes in insect life history [52] such as slower pre-adult development, reduced adult survival [53], decreased larval competitive success and adult fecundity [54] and decreases in body size [55], as shown for *Drosophila melanogaster*, but similar processes may occur also in beetles. Another possible explanation can be the use of artificial diet. Carabid beetles, like many invertebrates, derive energy from a variety of compounds, and a partially and/or fully artificial diet can significantly affect the insect development, in particular its mass and growth rate [56]. We cannot exclude that the artificial diet used in our culture lacks some elements or compounds necessary to achieve high body mass. Nevertheless, it did not prevent the successful culturing of the beetles through three generations, enabling us to separate out the actual genetically fixed adaptation from possible direct or maternal effects [57]. It should be stressed that despite the overall decrease in body mass in consecutive generations, the survival of the LM and LL beetles exposed to the insecticide was significantly and similarly higher than all other populations in all three generations.

## Conclusions

We confirmed that the effect of insecticides on the evolution of resistance in populations of the beneficial insect, the ground beetle *P. cupreus*, strongly depends on large-scale landscape characteristics. The small-fields landscape, with its higher spatio-temporal heterogeneity, is able to provide refuges that allow the beetles (and possibly other NTAs) to survive without developing any, presumably costly, resistance mechanisms. Understanding the significance of landscape structure for insect responses to pesticide pressures poses a key challenge in developing balanced pest control strategies and can be fundamental for forecasting the consequences of agricultural practices. NTAs, especially beneficial insects serving as natural pest enemies or pollinators, should be able not only to survive insecticide sprays, but also to provide ecosystem services. This requires maintenance of vital habitat conditions to prevent beneficial species from extinction or impairment. While increased resistance to pesticides among beneficial species can be considered a desired phenomenon, it has to be kept in mind that such adaptations are usually costly and may negatively affect the performance of resistant individuals, when other stress factors appear. Although the evolution of resistance is a complex process, influenced by many factors, we believe that our results emphasize the importance of landscape-scale management and pesticide use in agricultural management practices.

## Supporting information

**S1 Table. Proteus 110 OD (Bayer, Germany) specification.**
(DOCX)

## Acknowledgments

We would like to thank Renata Śliwińska, Paweł Dudzik and Edward Szmyd for their invaluable help in the field work.

## Author Contributions

**Conceptualization:** Grzegorz Sowa, Agnieszka J. Bednarska, Elżbieta Ziółkowska, Ryszard Laskowski.

**Data curation:** Grzegorz Sowa.

**Formal analysis:** Grzegorz Sowa.

**Funding acquisition:** Grzegorz Sowa.

**Investigation:** Grzegorz Sowa.

**Methodology:** Grzegorz Sowa, Agnieszka J. Bednarska, Elżbieta Ziółkowska, Ryszard Laskowski.

**Project administration:** Grzegorz Sowa.

**Resources:** Grzegorz Sowa.

**Validation:** Grzegorz Sowa.

**Visualization:** Grzegorz Sowa, Elżbieta Ziółkowska.

**Writing – original draft:** Grzegorz Sowa.

**Writing – review & editing:** Agnieszka J. Bednarska, Elżbieta Ziółkowska, Ryszard Laskowski.

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
