## [Decision Letter · Decision Letter 0]

20 Oct 2021

PONE-D-21-27829Homogeneity of agriculture landscape promotes insecticide resistance in the ground beetle Poecilus cupreusPLOS ONE

Dear Dr. Sowa,

Thank you for submitting your manuscript to PLOS ONE. After careful consideration, we feel that it has merit but does not fully meet PLOS ONE’s publication criteria as it currently stands. Therefore, we invite you to submit a revised version of the manuscript that addresses the points raised during the review process.

We look forward to receiving your revised manuscript.

Kind regards,

Ramzi Mansour

Academic Editor

PLOS ONE

Journal Requirements:

2. In your Methods section, please provide additional location information, including geographic coordinates of your field collection site if available.

a) You may seek permission from the original copyright holder of Figure 1 to publish the content specifically under the CC BY 4.0 license.  

Reviewers' comments:

Reviewer's Responses to Questions

**Comments to the Author**

1. Is the manuscript technically sound, and do the data support the conclusions?

Reviewer #1: Yes

2. Has the statistical analysis been performed appropriately and rigorously? 

Reviewer #1: Yes

3. Have the authors made all data underlying the findings in their manuscript fully available?

Reviewer #1: Yes

4. Is the manuscript presented in an intelligible fashion and written in standard English?

Reviewer #1: Yes

5. Review Comments to the Author

Reviewer #1: PONE-D-21-27829

Title: Homogeneity of agriculture landscape promotes insecticide resistance in the ground beetle Poecilus cupreus

Brief.

The manuscript was well written, and the methodology approach was appropriate for this study. Nonetheless, references were missed in the introduction section. In addition, part of the materials and methods sections is long. Although it is interesting, part of the information provided in the materials and methods sections should be moved to the discussion section. Suggestions and corrections were left in the general comments to improve the manuscript.

General comments

L27 – Please, ‘inheritable’ should be deleted from the sentence. Even though the sensitivity of P. cupreus was lower in the subsequent generation the inheritability of resistance of P. cupreus to Thiacloprid + Deltamethrin is not proved through these experiments and results.

L80 – Please, see the references Rodrigues et al. 2013 Response of different populations of seven lady beetle species to lambda-cyhalothrin with a record of resistance https://doi.org/10.1016/j.ecoenv.2013.06.014 and Torres et al. 2015 Lambda-Cyhalothrin Resistance in the Lady Beetle Eriopis connexa (Coleoptera: Coccinellidae) Confers Tolerance to Other Pyrethroids https://doi.org/10.1093/jee/tou035

L181, 183, … – Please, the scientific name of species starts the sentence with full name. This should be corrected in the entire manuscript.

L173 to 188 – The subsection is interesting from the point of view of details on the species studied. However, this subsection contains a lot of information that would not be appropriate for the materials and methods sections. Some of this information can be moved to the discussion section. Information about the species in previous studies detailed here will be important to explain the results found in this research.

L208 – Please, the commercial name of insecticides should be avoided in the manuscript because the commercial name of insecticide could be different in other countries. Instead, the name of insecticide active ingredients would be indicated to use in the manuscript.

L212 to 215 – The mode of action of each active ingredient (Thiacloprid + Deltamethrin) of insecticide (Proteus 110) should not be described in the materials and methods section. This type of information should be suppressed or moved to the introduction or discussion sections.

L215 to L220 – The reasons about the application history of specific insecticides presented here are interesting, but they are not suitable for the materials and methods section. Please, the sentences should be moved to the discussion section.

L236 to 239 – The sentence should be deleted. The criteria for the assessment of beetles have already been defined in the previous sentence.

L247 to 249 – Please, the sentence should be deleted from the statistical analysis subsection. Perhaps, the reference should be left in the sentence. However, the explanation why one was chosen than the other analysis is not appropriate in the materials and methods section. Although this information clarifies why one analysis is appropriate than the other it became a long manuscript with an excess of information.

Figure 4 – The legend of the x-axis was missed in figure 4. Please, the legend of the x-axis should be inserted in figure 4.

6. PLOS authors have the option to publish the peer review history of their article (what does this mean?). If published, this will include your full peer review and any attached files.

Reviewer #1: No

---

## [Author Response · Author response to Decision Letter 0]

26 Nov 2021

Response to Reviewer and Editor

We thank the Referee and Editor for the helpful comments. Below we provide point-by-point answers to the points raised by the Reviewer #1 and the Editor.

REVIEWER #1:

1. L27 – Please, ‘inheritable’ should be deleted from the sentence. Even though the sensitivity of P. cupreus was lower in the subsequent generation the inheritability of resistance of P. cupreus to Thiacloprid + Deltamethrin is not proved through these experiments and results.

We admit that we have not proven that any inheritable resistance mechanism has developed in the studied population, another possibility being the selection of more resistant individuals from the large populations inhabiting the studied areas. We have clarified this in the revised manuscript as follows: “The persistence of reduced sensitivity to Proteus 110 OD for two consecutive generations indicates that either the beetles have developed resistance to the insecticide or the chronic exposure to pesticides has led to the selection of more resistant individuals naturally present in the studied populations” (lines 23-26 in the revised paper without tracked changes).

2. L80 – Please, see the references Rodrigues et al. 2013 Response of different populations of seven lady beetle species to lambda-cyhalothrin with a record of resistance https://doi.org/10.1016/j.ecoenv.2013.06.014 and Torres et al. 2015 Lambda-Cyhalothrin Resistance in the Lady Beetle Eriopis connexa (Coleoptera: Coccinellidae) Confers Tolerance to Other Pyrethroids https://doi.org/10.1093/jee/tou035.

We are grateful to the Reviewer for pointing on these important publications, which are now cited in the revised manuscript (line 80 and 391) and have been added to the list of references (lines 549-555).

3. L181, 183, … – Please, the scientific name of species starts the sentence with full name. This should be corrected in the entire manuscript.

We have changed our wording across the manuscript as indicated by the Reviewer. 

4. L173 to 188 – The subsection is interesting from the point of view of details on the species studied. However, this subsection contains a lot of information that would not be appropriate for the materials and methods sections. Some of this information can be moved to the discussion section. Information about the species in previous studies detailed here will be important to explain the results found in this research.

We thought through this comment carefully, but would prefer to leave this sub-section as it was in the original version of the manuscript. In this sub-section, only information which allowed us to select this particular species for our research, determine the time of beetles collecting and select specific research sites (and crop) has been included (e.g., species numerous enough at all studies sites, disperse mainly by walking, easy to identify in the field, with a well-tested culturing procedure, one of the most common and dominant carabids found on arable land across Europe, including oilseed rape). Moreover, as the Reviewer stressed information detailed here is important to explain some of our results. We agree, that description of experimental species, its special characteristics and criteria for selection, does not necessary need to be a separate sub-section in manuscripts; the general methodological approach, including species description, often is a part of Introduction. We are not aware of any strict rules for placing the criteria for selecting species in the manuscript, but we feel it is better to have one consistent paragraph instead of covering species selection criteria in several places in the manuscript. Of course, if the Editor decides that this sub-section must be changed, we see no problem to include some information about the species in other places in in the manuscript, although Introduction seems to be better place at least for some information than Discussion.

5. L208 – Please, the commercial name of insecticides should be avoided in the manuscript because the commercial name of insecticide could be different in other countries. Instead, the name of insecticide active ingredients would be indicated to use in the manuscript.

The reviewer is right that reporting the commercial name of an insecticide does not guarantee exactly same composition in different countries, but in this case we cannot refer to the active ingredients only because the purpose of the experiment was to test the actual product used by farmers (i.e., active ingredients + additives), not the specific active ingredients. Please note, however, that the exact contents of the active ingredients in the product are reported in the manuscript (line 218-219). Moreover, although exact concentrations of all product ingredients except the insecticides themselves may differ slightly, the commercial name usually means a mixture of specific chemicals. Hence, we prefer to keep in the manuscript the commercial name of the product used.

6. L212 to 215 – The mode of action of each active ingredient (Thiacloprid + Deltamethrin) of insecticide (Proteus 110) should not be described in the materials and methods section. This type of information should be suppressed or moved to the introduction or discussion sections.

As suggested by the Reviewer, the sentence describing the modes of action of both active ingredients has been moved to the Discussion. We also added a few sentences stressing the necessity of using actual formulations in pesticide testing. The section reads now: “Proteus 110 OD is the plant protection product containing two active ingredients, thiacloprid which belongs to neonicotinoids and affects insect nervous system by stimulating nicotinic acetylcholine receptors, and deltamethrin which is a pyrethroid preventing the closure of the voltage-gated sodium channels in the axonal membranes resulting in dysfunction of spiracles – insect death is caused by overdrying. However, the ultimate effect of commercial formulations depends not only on the active ingredients but also adjuvants/co-formulants, which may affect the efficacy of penetration of the active ingredients into insect bodies or the persistence of an insecticide in field conditions [45]. Hence, other formulations with exactly same active ingredients may have somewhat different effects than those reported herein. (lines 369-377).

7. L215 to L220 – The reasons about the application history of specific insecticides presented here are interesting, but they are not suitable for the materials and methods section. Please, the sentences should be moved to the discussion section.

The indicated sentence has been moved to the Discussion section (lines 377-380).

8. L236 to 239 – The sentence should be deleted. The criteria for the assessment of beetles have already been defined in the previous sentence.

Done.

9. L247 to 249 – Please, the sentence should be deleted from the statistical analysis subsection. Perhaps, the reference should be left in the sentence. However, the explanation why one was chosen than the other analysis is not appropriate in the materials and methods section. Although this information clarifies why one analysis is appropriate than the other it became a long manuscript with an excess of information.

Although indeed this sentence may appear not crucial as this can be considered a general knowledge of statistics, the Reviewer admitted himself that “this information clarifies why one analysis is [more] appropriate than the other”. Because we are convinced that in this particular case it is important to clarify why this specific test was used, we prefer to keep this sentence in the manuscript. These are just two lines of text so we don’t think that shortening the manuscript by 30 words is worth sacrificing this information. If, however, the Editor and the Reviewer decide otherwise, we can, of course, make the necessary correction. 

FIGURES

Figure 4 – The legend of the x-axis was missed in figure 4. Please, the legend of the x-axis should be inserted in figure 4.

The x-axis legend has been added (“Habitat type”) and we have also unified with other figures the descriptions of the three plots relating to the three consecutive generations (red uppercase letters “P”, “F1”, “F2”).

Response to Editor

The manuscript has been revised to meet PLOS ONES’s style requirements as provided in the above mentioned templates.

2. In your Methods section, please provide additional location information, including geographic coordinates of your field collection site if available.

Information on the coordinates of all twelve study sites has been included as Table 2.

The explanation of the permits obtained has been added and the text now reads: “The authorisation for capture, transport and keeping the beetles was granted by the Regional Directorate for Environmental Protection in Kraków, Poland, document no. OP-I.6401.128.2017.MMr, and by the Regional Directorate for Environmental Protection in Poznań, Poland, document no. WPN-II.6401.83.2017.AG.2.” (lines 198-201).

Yes, we confirm that repository information for our data will be provided upon acceptance of our manuscript

The above mentioned figure is neither in whole nor in part a satellite image. The upper part of the figure showing location of the two study landscapes in the Wielkopolska province and the location of the Wielkopolska province in Poland was generated using data from the National Register of Boundaries in Poland which is provided free of charge (via the service www.geoportal.gov.pl) and can be used for any purpose according to the amendments to the Surveying and Cartographic Act from April 16 2020. The land-cover maps of our study landscapes shown at the bottom of the figure are simplified versions of very detailed land cover maps generated by us in a step-by step process using the methodology described in Ziółkowska et al. (2021) based on two types of source data: (1) the National Database of Topographic Objects (BDOT) providing the land cover and land use information at the scale of 1:10.000, and (2) the Land Parcel Identification System (LPIS) providing information on the type of cultivated crops (see Methods section). We are the copyright holders of these products. However, the source data (BDOT database and data on parcels - Land and building register) are also provided free of charge (via the service www.geoportal.gov.pl) and can be used for any purpose according to the amendments to the Surveying and Cartographic Act from April 16 2020. Therefore, the entire figure may be published under the Creative Commons Attribution License (CC BY 4.0).

In addition, to be more clear, we have now added the following explanation to the main text of the manuscript: “Raster land-cover maps (resolution of 1 m2) for the landscapes were generated in a step-by-step process according to the methodology described in [33] by combining data from: (1) the National Database of Topographic Objects (BDOT) providing the land cover and land use information at the scale of 1:10.000, and (2) the Land Parcel Identification System (LPIS) providing information on the type of cultivated crops.” [lines 145-149]

We also provided some additional explanation in the Figure 1 caption:

“Land-cover maps were generated according to methodology described by Ziółkowska et al. [33], see Methods section for more details.” [lines 139-140].

---

## [Decision Letter · Decision Letter 1]

17 Dec 2021

PONE-D-21-27829R1Homogeneity of agriculture landscape promotes insecticide resistance in the ground beetle Poecilus cupreusPLOS ONE

Dear Dr. Sowa,

Thank you for submitting your manuscript to PLOS ONE. After careful consideration, we feel that it has merit but does not fully meet PLOS ONE’s publication criteria as it currently stands. Therefore, we invite you to submit a revised version of the manuscript that addresses the points raised during the review process.

We look forward to receiving your revised manuscript.

Kind regards,

Ramzi Mansour

Academic Editor

PLOS ONE

Reviewers' comments:

Reviewer's Responses to Questions

**Comments to the Author**

1. If the authors have adequately addressed your comments raised in a previous round of review and you feel that this manuscript is now acceptable for publication, you may indicate that here to bypass the “Comments to the Author” section, enter your conflict of interest statement in the “Confidential to Editor” section, and submit your "Accept" recommendation.

Reviewer #1: (No Response)

2. Is the manuscript technically sound, and do the data support the conclusions?

Reviewer #1: Yes

3. Has the statistical analysis been performed appropriately and rigorously? 

Reviewer #1: Yes

4. Have the authors made all data underlying the findings in their manuscript fully available?

Reviewer #1: Yes

5. Is the manuscript presented in an intelligible fashion and written in standard English?

Reviewer #1: (No Response)

6. Review Comments to the Author

Reviewer #1: #PONE-D-21-27829R1

Title: Homogeneity of agriculture landscape promotes insecticide resistance in the ground beetle Poecilus cupreus

Brief.

The manuscript has been improved after revisions. However, some critical points of the manuscript should be a better-clarified regarding the results found in the experiments and terms used throughout the manuscript. Furthermore, a broad interval confidence interval found for the LT50 values of the LA and LL habitat beetles should be appropriately discussed in the manuscript. Suggestions and corrections were left in the general comments to improve the manuscript.

General comments

L19, 24, 105, 119, 214, 227, 235, 284, 292, 314, 326, 362, 367, 369, and 402 - The reason given in the section of the response to the reviewers by the author for keeping the insecticide trade name Proteus 110 OD in the manuscript is conflicting because the insecticide Proteus 110 OD is a mixture of ingredient actives thiacloprid + deltamethrin. Several products with ingredient actives mixture have been released in the last years. What were the additives tested in the experiment? Oil dispersion (nonpolar) allows break plant barriers such as the wax layer of leaves plants (nonpolar) to lead the ingredient active improving an insecticide’s uptake by the plant. Acetone is an adjuvant that should not have to act in the beetle. The suggestion to change the trade name of Proteus 110 OD to its active ingredient was based on making it internationally standardized to identify it other countries exclusively through the name of ingredient active. However, decisions can be made by consensus of the author and editor.

L81 to 82, 86, 95, 171, 236, 268, 269, 352, 395, 397, 421, 465 and 470 – Please, all references used in the sentence referred to insecticides. Thus, pesticides should be changed to insecticides. Pesticide is a broad term that includes various products to kill different types of organisms, insecticide is a term designated for products that act specifically on insects. Please, the correct term should be reviewed in the entire manuscript.

L345 to 348 – The broad confidence intervals to the LT50 values for insecticide-treated beetles from the LM and LL habitat types and acetone-treated beetles from the control habitats (S0-A and L0-A, Fig 4) should be discussed in the discussion section. Why were only beetles from these areas had a broad confidence interval in the different generations' studies (P, F1, and F2)?

L339 and 351 - How was it determined that the beetle was tolerant to insecticides? Would not this be an evolution of the beetle's resistance to insecticides? The term used to describe the results found in the experiment should be clearer.

L360 to 362 - The first sentence of the discussion section contradicts the results described in the materials and methods section, see comments above in lines 339 and 351. In materials and methods, the tolerance of the beetle to insecticides has been described. However, the discussion concerns the development of insecticide resistance in the beetle. Please, the terms used in the materials and methods section and the discussion should be clearer.

L377 to 380 - The sentence is confusing as a massive number of studies have used both commercial formulations and unique active ingredients. Please, the sentence should be rephrased or deleted from the discussion section.

L408 to 409 – There was no found tolerance as well as the use of pesticides in this study mentioned in the sentence. Schoonees and Giliomee (1982) found an increase of resistance of two strains of parasitoids to the insecticides in different localities where the application of insecticide was intensified through the years.

7. PLOS authors have the option to publish the peer review history of their article (what does this mean?). If published, this will include your full peer review and any attached files.

Reviewer #1: No

---

## [Author Response · Author response to Decision Letter 1]

10 Jan 2022

Response to Reviewer

We thank the Referee for the helpful comments. Below we provide point-by-point answers to the points raised by the Reviewer #1.

REVIEWER #1:

1. L19, 24, 105, 119, 214, 227, 235, 284, 292, 314, 326, 362, 367, 369, and 402 - The reason given in the section of the response to the reviewers by the author for keeping the insecticide trade name Proteus 110 OD in the manuscript is conflicting because the insecticide Proteus 110 OD is a mixture of ingredient actives thiacloprid + deltamethrin. Several products with ingredient actives mixture have been released in the last years. What were the additives tested in the experiment? Oil dispersion (nonpolar) allows break plant barriers such as the wax layer of leaves plants (nonpolar) to lead the ingredient active improving an insecticide’s uptake by the plant. Acetone is an adjuvant that should not have to act in the beetle. The suggestion to change the trade name of Proteus 110 OD to its active ingredient was based on making it internationally standardized to identify it other countries exclusively through the name of ingredient active. However, decisions can be made by consensus of the author and editor.

We still agree with the Reviewer on the fact that when reporting the commercial name of an insecticide, one can expect different composition in different countries. However, we stand by our arguments that providing information only on active ingredients, without reporting commercial name, may be confusing. By giving the full name of the product, together with the exact contents of the active ingredients in the product and the country of purchase (lines 212-218 in the revised paper without tracked changes), we avoid any doubts whether the toxicity of the active ingredients alone could be potentially different from that of the mixture (i.e. the commercially available product). The purpose of the experiment was to test the actual product used by farmers (i.e., active ingredients + additives), not the specific active ingredients, as we wanted to reflect as closely as possible the real situation in the field. To make it unambiguously clear what the composition of the product used was, we have included, as Supporting information, a table with product specification as provided by the manufacturer (Bayer, Germany) (S1 Table). Additional text in lines 217-218 in the revised manuscript was also added: “(see Table S1 in Supplementary materials for full product specification)”.

2. L81 to 82, 86, 95, 171, 236, 268, 269, 352, 395, 397, 421, 465 and 470 – Please, all references used in the sentence referred to insecticides. Thus, pesticides should be changed to insecticides. Pesticide is a broad term that includes various products to kill different types of organisms, insecticide is a term designated for products that act specifically on insects. Please, the correct term should be reviewed in the entire manuscript.

Of course we agree with the Reviewer that pesticide is a broad term that includes various products (e.g. insecticides, herbicides, fungicides) to kill different types of organisms, whereas insecticides act specifically on insects. We have now changed the term “pesticide” into “insecticide” wherever we refer to our laboratory test results on Proteus 110 OD, which is the insecticide (lines 236, 268, 269, 352, now lines 234, 265, 266, 345). The changes were also done in lines 81-82 and 86 (now lines 79-80 and 84): although the references we used are not specific to insecticides (they relate to resistance to xenobiotics in general), we agree that for consistency with the earlier sentence, in which “insecticides” were mentioned, the same term should be used in the indicated sentences. In line 94, the term “pesticide” was mentioned because the use of other products then insecticides (e.g. herbicides, fungicides), even though they are not directly harmful to ground beetles, still may indirectly influence their survival, for example through the impact on their habitat and/or food source (e.g., disappearance of plant food). Therefore, the use of the term “pesticide” is appropriate in this context. A similar situation arises in line 171 (now line 169 in the revised paper without tracked changes) where the term “pesticide” is used to emphasize that the populations of the two habitats mentioned (S0 and L0) were not exposed to any potentially toxic substances in the field. The use of the term “insecticides” would imply that other types of treatments could have been applied there. A similar justification for using a term “pesticide” applies to line 421 (now line 411). Regarding lines 465 and 470 (now lines 455 and 460), we decided to keep the original terminology, as we draw general conclusions here on either significance of landscape for insect response (not resistance) to pressures from different pesticides used in that landscape (line 465, now line 455) or resistance of beneficial species (not specifically insects) to pesticides (line 470, now line 460).

3. L345 to 348 – The broad confidence intervals to the LT50 values for insecticide-treated beetles from the LM and LL habitat types and acetone-treated beetles from the control habitats (S0-A and L0-A, Fig 4) should be discussed in the discussion section. Why were only beetles from these areas had a broad confidence interval in the different generations' studies (P, F1, and F2)?

The confidence intervals to the LT50 were calculated fusing Litchfield’s method (lines 256 – 258 in the previous version of the manuscript) which is a method for rapid graphic solution of time-per cent effect curves. We have now checked that confidence intervals are much narrower when they are calculated for medians (LT50 is the median lethal time). To give an example, for insecticide-treated beetles from LL habitat types in F2 generation, with LT50 of 15.23 days, the confidence interval calculated using Litchfield’s method is between 5.67 and 40.92 days, whereas the confidence interval for the median of 15.23 days is between 12.23 and 16.23 days (based on equations on https://www.statology.org/confidence-interval-for-median/).

However, following the Reviewer’s remark about the broad confidence intervals to the LT50 values we concluded that we unnecessarily confused the reader suggesting that confidence intervals were used to compare the mortality rates between the habitat types. In fact, as stressed in table 3, we compared the whole survival curves using Wilcoxon test (“Groups with the same lowercase letter do not differ significantly at p ��0.05 in terms of survival (pairwise comparison of survival curves, Wilcoxon test)”, now lines 299). Because of that, we decided to replace the Figure 4 with a similar figure, but instead confidence intervals, standard errors for LT50 values are now presented. Thus, all sentences refereeing to confidence intervals in Statistical analysis and results describing differences comparison of LT50s between habitat types based on confidence intervals overlap have been removed from the revised version of the manuscript.

4. L339 and 351 - How was it determined that the beetle was tolerant to insecticides? Would not this be an evolution of the beetle's resistance to insecticides? The term used to describe the results found in the experiment should be clearer.

Each individual beetle from each habitat type had its own survival time after insecticide treatment (i.e., its own insecticide sensitivity = insecticide tolerance) and we showed that: “as in the P generation, the comparison among insecticide-treated beetles from different habitat types revealed statistically significant difference in survival rates of beetles in both the F1 and F2 generations (p < 0.001 for F1, p < 0.03 for F2, Fig 3)” (now lines 332 - 335) and “The increased tolerance to the insecticide found in beetles from the P generation from the LM and LL habitats was still present in both next generations F1 and F2, as insecticide-treated beetles from the LM and LL habitats survived significantly better than insecticide-treated beetles from all other habitat types (Fig 3, Table 3), but significantly worse than acetone-treated beetles from the control habitats (S0-A and L0-A; Table 3)”. Thus, because similar results of increased tolerance to the insecticide were found in beetles from the LM and LL habitats over three generations (P, F1 and F2), the results suggest the evolution of resistance. Results and Discussion sections are, however, separated in the manuscript, so we do not interpret our results in Results section. We assumed that the Discussion section is designed to tell what the results mean. Thus, we would prefer to keep our original writing and first present the results on increased tolerance to the insecticide in beetles from the LM and LL habitats over three generations (P, F1 and F2) and then conclude that such results suggest the evolution of resistance. Nevertheless, if the Editor decides otherwise, we don't see a problem with adding following summary/conclusion sentence to the Results section: “Thus, the results suggest that the effect of insecticides on the evolution of resistance in populations of P. cupreus strongly depends on large-scale landscape characteristics.”.

5. L360 to 362 - The first sentence of the discussion section contradicts the results described in the materials and methods section, see comments above in lines 339 and 351. In materials and methods, the tolerance of the beetle to insecticides has been described. However, the discussion concerns the development of insecticide resistance in the beetle. Please, the terms used in the materials and methods section and the discussion should be clearer.

Yes, in the Material and Methods section we described the methods used to study the tolerance of the beetles to insecticide (in other words, we studied susceptibility of beetles to pesticide). This tolerance was studied for three generations to sort out possible temporary effects through direct selection of the most resistant individuals collected from the field right after the spraying from parental effect and/or possible genetically fixed resistance to insecticides (as stressed already in the Introduction, lines 114 -118 in the revised paper without tracked changes). As we stressed in the Discussion section, the successful culturing of the beetles through three generations enabled us to separate the actual genetically fixed adaptation from possible direct or maternal effects (lines 443 - 445). Thus, we don’t see any contradiction between Methods (testing tolerance of beetles to insecticides in generation P, F1 and F2) and Discussion sections (drawing conclusions from obtaining similar results on tolerance of beetles to insecticide in all three generations). Please see also our answer to the previous comment as well as to the last comment. 

6. L377 to 380 - The sentence is confusing as a massive number of studies have used both commercial formulations and unique active ingredients. Please, the sentence should be rephrased or deleted from the discussion section.

As suggested the mentioned sentence has been deleted form the discussion section.

7. L408 to 409 – There was no found tolerance as well as the use of pesticides in this study mentioned in the sentence. Schoonees and Giliomee (1982) found an increase of resistance of two strains of parasitoids to the insecticides in different localities where the application of insecticide was intensified through the years.

Schoonees and Giliomee (1982) evaluated the toxicity of methidathion (organophosphorus insecticide) to two strains of Aphytis africanus and Comperiella bifasciata, both parasitoids of Aonidiella aurantii in South Africa. They found that different geographic strains varied; one of them, which used to receive three to four sprays of organophosphorus insecticides per year over many years, was much more tolerant to methidathion than the one, which received only one spray per year. To be precise, the authors used the word “susceptibility” to check difference of the studied species between localities, but being more susceptible to something means less tolerance to it. In the Methods they described how this “susceptibility” was measured and expressed (survival was checked in contact toxicity test and LC50 was calculated). In the Results they described differences in susceptibility between the two strains (collected from different localities) and calculated “ resistance factor”, which is LC50 of one strain divided by LC50 of the second strain. In the Discussion section, the authors concluded about “the greater resistance to methidathion in A. africanus and C. bifasciata from Letsitele as compared with those from Mooinooi and Zebediela” which simply means that A. africanus and C. bifasciata from Letsitele had higher LC50 (were less susceptible to methidathion = were more tolerant to methidathion) as compared with those from Mooinooi and Zebediela.

The paper by Schoonees and Giliomee (1982) proves that organisms subjected to higher number of insecticide sprays are able to develop tolerance to the most commonly used product (=are more resistant, and the word “tolerance” has been now replaced by the “resistance”). We agree that the paper does not directly answer the question of whether landscape structure is one of the causes of resistance (=being more tolerant to methidathion). However, the described situation (higher number of sprays) may occur more often in homogenised landscapes where the insects inhabiting them have limited or no access to a refuges. With this in mind, we consider the work by Schoonees and Giliomee (1982) as a good example that more frequent contact with insecticides promotes the development of resistance.

ADDITIONAL COMMENT

In Table 3, we made a typo error for the LT50 values of beetles from P generation and LM habitat. Instead of 21.25 it should be 23.26.

---

## [Decision Letter · Decision Letter 2]

20 Jan 2022

PONE-D-21-27829R2

Homogeneity of agriculture landscape promotes insecticide resistance in the ground beetle Poecilus cupreus

PLOS ONE

Dear Dr. Sowa,

Thank you for submitting your manuscript to PLOS ONE. After careful consideration, we have decided that your manuscript does not meet our criteria for publication and must therefore be rejected.

For the second consecutive time, the authors clearly failed to address the expert Reviewer's comments and suggestions and provided confusing responses. Additionally, as stated by the expert Reviewer, there is a confusion between (wrong use and interpretation) the important scientific terms "resistance" and "tolerance" in the manuscript and wrong reedits from previous scientific articles. Accordingly, this manuscript could no longer be considered for publication in PLOS ONE. All comments of the expert Reviewer are stated below.

I am sorry that we cannot be more positive on this occasion, but hope that you appreciate the reasons for this decision.

Yours sincerely,

Ramzi Mansour

Academic Editor

PLOS ONE

Reviewers' comments:

Reviewer's Responses to Questions

**Comments to the Author**

1. If the authors have adequately addressed your comments raised in a previous round of review and you feel that this manuscript is now acceptable for publication, you may indicate that here to bypass the “Comments to the Author” section, enter your conflict of interest statement in the “Confidential to Editor” section, and submit your "Accept" recommendation.

Reviewer #1: (No Response)

2. Is the manuscript technically sound, and do the data support the conclusions?

Reviewer #1: (No Response)

3. Has the statistical analysis been performed appropriately and rigorously? 

Reviewer #1: (No Response)

4. Have the authors made all data underlying the findings in their manuscript fully available?

Reviewer #1: (No Response)

5. Is the manuscript presented in an intelligible fashion and written in standard English?

Reviewer #1: (No Response)

6. Review Comments to the Author

Reviewer #1: # PONE-D-21-27829R2

Title: Homogeneity of agriculture landscape promotes insecticide resistance in the ground beetle Poecilus cupreus

Brief.

The manuscript was partially revised following suggestions from the reviewers. Furthermore, even after revising the terms used in the manuscript, the author chose to keep the wrong term and justify the error by reediting the term used in another manuscript.

General comments

The comments about lines 339 and 351 and 360 to 362 is not suggesting any discuss. The terms “tolerance” and “resistance” in results and discussion section was not clear and it remain unclear. Briefly, resistance may be defined as “a heritable change in the sensitivity of a pest population that is reflected in the repeated failure of a product to achieve the expected level of control when used according to the label recommendation for that pest species” (IRAC, 2022 [https://irac-online.org/about/resistance/]). In contrast to resistance, insecticide tolerance is a natural tendency and not a result of selection pressure and following Tabashnik (1991) insecticide resistance can be viewed as a threshold trait, with tolerance as the underlying continuous variable (Finney 1971, Falconer 1981, Via 1986, Tabashnik & Cushing 1989). Thus, the term should be rethought, because if beetles have been exposed to active ingredients at the collect site and exposed in the generations P, F1 and F2 generations, would they be resistant or tolerant? There is an expectation supported by results found that beetles are not tolerant, but individuals resistant. This would all change the way approach this term in the sections. Again, a discussion of this in any section is not recommended.

Bruce E. Tabashnik, 1991. Deterrnining the Mode of Inheritance of Pesticide Resistance with Backcross Experiments. Journal of Economic Entomology. V. 84, no. 3, p.703-712.

The explanation provide by authors of manuscript about lines 408 to 409 in the response to reviewers is completely wrong and they are reediting the terms “susceptibility and resistance” used correctly by Schoonees and Giliomee (1982).

7. PLOS authors have the option to publish the peer review history of their article (what does this mean?). If published, this will include your full peer review and any attached files.

Reviewer #1: No

- - - - -

---

## [Author Response · Author response to Decision Letter 2]

8 Feb 2022

Response to the Reviewer

We acknowledge receiving the decision rejecting our manuscript based on the review, but we fundamentally do not agree with the opinion expressed by the Reviewer and, consequently, the decision made by the Academic Editor. The Reviewer had only one comment, relating to the use of terms “tolerance” and “resistance” in our manuscript. Below please find the comment itself (in blue italics, indented) followed by our rebuttal.

Reviewer #1: # PONE-D-21-27829R2

Title: Homogeneity of agriculture landscape promotes insecticide resistance in the ground beetle Poecilus cupreus

Brief.

The manuscript was partially revised following suggestions from the reviewers. Furthermore, even after revising the terms used in the manuscript, the author chose to keep the wrong term and justify the error by reediting the term used in another

manuscript.

General comments

The comments about lines 339 and 351 and 360 to 362 is not suggesting any discuss. The terms “tolerance” and “resistance” in results and discussion section was not clear and it remain unclear. Briefly, resistance may be defined as “a heritable change in the sensitivity of a pest population that is reflected in the repeated failure of a product to achieve the expected level of control when used according to the label recommendation for that pest species” (IRAC, 2022 [https://iraconline.org/about/resistance/]). In contrast to resistance, insecticide tolerance is a natural tendency and not a result of selection pressure and following Tabashnik (1991) insecticide resistance can be viewed as a threshold trait, with tolerance as the underlying continuous variable (Finney 1971, Falconer 1981, Via 1986, Tabashnik & Cushing 1989). Thus, the term should be rethought, because if beetles have been exposed to active ingredients at the collect site and exposed in the generations P, F1 and F2 generations, would they be resistant or tolerant? There is an expectation supported by results found that beetles are not tolerant, but individuals resistant. This would all change the way approach this term in the sections. Again, a discussion of this in any section is not recommended.

Bruce E. Tabashnik, 1991. Deterrnining the Mode of Inheritance of Pesticide Resistance with Backcross Experiments. Journal of Economic Entomology. V. 84, no.

3, p.703-712.

The explanation provide by authors of manuscript about lines 408 to 409 in the response to reviewers is completely wrong and they are reediting the terms

“susceptibility and resistance” used correctly by Schoonees and Giliomee (1982).

We do not agree with the opinion expressed by the Reviewer and with the decision of the Academic Editor rejecting our manuscript. First of all, it should be noted that the Reviewer did not indicate any flaws in our study, data analysis or the manuscript itself except of contesting the use of just two terms: “tolerance” and “resistance”. In our opinion this might be, at the very best, the subject of discussion on the semantics but does not justify rejecting the manuscript. The questioned terms do not have such a strict scientific meaning as the Reviewer tries to imply. The meaning of the terms is defined in different ways, depending on the need, as shown in the examples below. Although some authors can define them in a way indicated by the Reviewer, this does not mean that such an understanding is the only one and obligatory. More importantly, in the revised version and in our previous Response to the Reviewer we explained very clearly how both terms are understood (defined) in our manuscript and why we decided to use both. Briefly, by testing tolerance to a pesticide (by estimating LC50 and comparing survival curves) throughout three generations we could conclude that most probably some of the tested populations exhibited genetically fixed resistance. This is exactly in accordance with the definition proposed by Tabashnik and Johnson in the “Handbook of Biological Control” (by the way, one of the authors, B. E. Tabashnik, is the very same author as the one cited by the Reviewer!):

Examples of the use of terms “resistance” and “tolerance”

“Pesticide resistance is a genetically based, statistically significant increase in the ability of a population to tolerate one or more pesticides. In most cases, resistance is documented with laboratory bioassays showing that a population with a history of extensive exposure to pesticides has a significantly greater LC50 or LD50 […] compared with a conspecific population that has had less exposure to pesticides. One can also document resistance by showing that treatment with a fixed concentration or dose causes significant differences in mortality among conspecific populations […]. Evidence of significant increase through time within a population in LC50, LD50, or survival in response to a fixed concentration or dose provides more direct documentation of resistance.” (from “Evolution of Pesticide Resistance in Natural Enemies” by B. E. Tabashnik, M. W. Johnson, in “Handbook of Biological Control”, 1999).

In turn, EPA defines tolerance in a very precise way for a very specific purpose: “A tolerance is the EPA established maximum residue level of a specific pesticide chemical that is permitted in or on a specific human or animal food in the United States”.

In pharmacology “Tolerance is a decrease in response to a drug that is used repeatedly. Resistance is development of the ability to withstand the previously destructive effect of a drug by microorganisms or tumor cells”.

Referring to pesticide resistance and tolerance: “The chemical arsenal we have developed in an attempt to rid our homes of rodents and our crops of insects is losing its power. We have simply caused pest populations to evolve, unintentionally applying artificial selection in the form of pesticides. Individuals with a higher tolerance for our poisons survive and breed, and soon resistant individuals outnumber the ones we can control.”

(https://www.pbs.org/wgbh/evolution/library/10/1/l_101_02.html)

Please note also that the English language thesaurus reports “resistance” and “resilience”

(together with “strength” and “toughness”) as the closest synonyms of “tolerance”.

Moreover, the Reviewer’s statement about the definition of resistance (“resistance may be defined as “a heritable change in the sensitivity of a pest population that is reflected in the repeated failure of a product to achieve the expected level of control when used according to the label recommendation for that pest species”) is completely irrelevant to our studies as we did not study effectiveness of any product against pests but rather the combined effect of infield pesticide applications and landscape structure on the resistance of a beneficial carabid. In our earlier response to the Reviewer, we provided a detailed answer to the Reviewer’s comments and explained that only “because similar results of increased tolerance to the insecticide were found in beetles from the LM and LL habitats over three generations (P, F1 and F2), the results suggest the evolution of resistance” (in terms of a heritable change in the sensitivity), as by doing the study on three generations of beetles we could “separate the actual genetically fixed adaptation (=heritable resistance) from possible direct or maternal effects”. We also explained that the increased tolerance to an insecticide means the decreased sensitivity to that insecticide. Even if this is not exactly the same as the Reviewer's reasoning, we are convinced that this is, at the best, a field for polemics rather than a basis for rejecting the manuscript. Indeed, if the Editor prefers to have the term “less sensitive” instead of “tolerant” (although tolerant means simply less sensitive), we do not see a problem with such a rewording.

We completely do not agree with the Reviewer’s statement that “The explanation provide by authors of manuscript about lines 408 to 409 in the response to reviewers is completely wrong and they are reediting the terms “susceptibility and resistance” used correctly by Schoonees and Giliomee (1982).” We read the paper carefully and would like to know where exactly we distorted the meaning of the words.

It must be stressed that the decision to reject our manuscript was based on the opinion of just one reviewer who apparently did not find any methodological flaws in our study or data analysis and simply has different opinion on the meaning of just two terms used in the manuscript. Also, in the first review no problem with the definition of the terms was mentioned by the Reviewer; nevertheless, the terms have been precisely defined in the revised version and should not bring any doubts about their meaning in the study.

---

## [Editor Report · Decision Letter 3]

22 Mar 2022

Homogeneity of agriculture landscape promotes insecticide resistance in the ground beetle Poecilus cupreus

PONE-D-21-27829R3

Dear Dr. Sowa

We’re pleased to inform you that your manuscript has been judged scientifically suitable for publication and will be formally accepted for publication once it meets all outstanding technical requirements.

Kind regards,

Tuneera Bhadauria, Ph.D.

Academic Editor

PLOS ONE

Additional Editor Comments (optional):

I believe they are extremely obvious in the text after reading the amended manuscript and the authors' responses to the reviewers' questions about the use of the phrases tolerance and resistance. I can explain it in the following context, as stated by the authors. I believe that the small-field environment, with its enhanced spatiotemporal variety, enables additional niches for beetles (and presumably other NTAs) to exist in while avoiding the development of insect resistance.In landscapes dominated by wide fields and ecosystem homogenization through mono cropping and recurrent periods of insecticide spraying, beetles are unable to move into fields with heterogenous cropping practices. This raises the chances of beetles being exposed to the insecticidal spray on a regular basis, developing tolerance to it over time, and finally becoming resistant to it.

I propose that the paper be approved for publishing in the journal because the authors have responded well to all of the additional clarifications, comments, and suggestions made by reviewers, incorporating them into the text as and where needed.
---

## [Editor Report · Acceptance letter]

18 Apr 2022

PONE-D-21-27829R3 

Homogeneity of agriculture landscape promotes insecticide resistance in the ground beetle *Poecilus cupreus*

Dear Dr. Sowa:

I'm pleased to inform you that your manuscript has been deemed suitable for publication in PLOS ONE. Congratulations! Your manuscript is now with our production department. 

Kind regards, 

on behalf of

Dr. Tunira Bhadauria 

Academic Editor

PLOS ONE